# Dental Pulp-Derived Mesenchymal Stem Cells for Modeling Genetic Disorders

**DOI:** 10.3390/ijms22052269

**Published:** 2021-02-25

**Authors:** Keiji Masuda, Xu Han, Hiroki Kato, Hiroshi Sato, Yu Zhang, Xiao Sun, Yuta Hirofuji, Haruyoshi Yamaza, Aya Yamada, Satoshi Fukumoto

**Affiliations:** 1Section of Oral Medicine for Children, Division of Oral Health, Growth and Development, Faculty of Dental Science, Kyushu University, Maidashi 3-1-1, Higashi-Ku, Fukuoka 812-8582, Japan; kan@dent.kyushu-u.ac.jp (X.H.); hisato@dent.kyushu-u.ac.jp (H.S.); zhangyu@dent.kyushu-u.ac.jp (Y.Z.); sunxiao1988@dent.kyushu-u.ac.jp (X.S.); hirofuji@dent.kyushu-u.ac.jp (Y.H.); hyamaza@dent.kyushu-u.ac.jp (H.Y.); 2Department of Molecular Cell Biology and Oral Anatomy, Graduate School of Dental Science, Kyushu University, Maidashi 3-1-1, Higashi-Ku, Fukuoka 812-8582, Japan; kato@dent.kyushu-u.ac.jp; 3Division of Pediatric Dentistry, Department of Oral Health and Development Sciences, Graduate School of Dentistry, Tohoku University, 4-1 Seiryo-machi, Aoba-ku, Sendai, Miyagi 980-8577, Japan; yamada-a@dent.tohoku.ac.jp

**Keywords:** dental pulp stem cells, disease model, mesenchymal stem cells, stem cells from human exfoliated deciduous teeth

## Abstract

A subpopulation of mesenchymal stem cells, developmentally derived from multipotent neural crest cells that form multiple facial tissues, resides within the dental pulp of human teeth. These stem cells show high proliferative capacity in vitro and are multipotent, including adipogenic, myogenic, osteogenic, chondrogenic, and neurogenic potential. Teeth containing viable cells are harvested via minimally invasive procedures, based on various clinical diagnoses, but then usually discarded as medical waste, indicating the relatively low ethical considerations to reuse these cells for medical applications. Previous studies have demonstrated that stem cells derived from healthy subjects are an excellent source for cell-based medicine, tissue regeneration, and bioengineering. Furthermore, stem cells donated by patients affected by genetic disorders can serve as in vitro models of disease-specific genetic variants, indicating additional applications of these stem cells with high plasticity. This review discusses the benefits, limitations, and perspectives of patient-derived dental pulp stem cells as alternatives that may complement other excellent, yet incomplete stem cell models, such as induced pluripotent stem cells, together with our recent data.

## 1. Introduction

Multipotent stem cells of mesenchymal origin are widespread in the postnatal connective tissues [1,2,3]. Non-hematopoietic multipotent stromal cells were first isolated from bone marrow and described as colony-forming unit-fibroblasts with clonogenic proliferation in vitro [4,5]. Stem cells with similar characteristics have also been identified in other tissues, including adipose tissue, liver, and cord blood, and are commonly referred to as mesenchymal stem cells (MSCs) [1,2,3,6]. They are involved in several physiological functions, including tissue homeostasis, turnover, and native regeneration [2]. Additionally, in vitro culture can induce osteogenic, chondrogenic, adipogenic, myogenic, and neurogenic plasticity in MSCs. They also exhibit lineage-committed and tissue-specific differentiation after transplantation in vivo. Furthermore, it has been shown that they may also modulate the immune system on systemic administration in the recipients [2,7,8]. Thus, MSCs play an important role in the developmental modeling and regeneration of tissues and are key mediators of cell-based therapy of damaged tissues. However, there are still many concerns to be resolved, including the selection of reliable cell sources to safely extract sufficient MSCs from the human body for medical research and application.

MSCs were reported to be present in the human teeth and supporting tissues, and these have been reported to be promising sources of MSCs [9,10,11]. These “Dental MSCs” are divided into several subpopulations, depending on their anatomical and histological origins. Mature deciduous and permanent teeth contain MSCs in their pulp and periodontal ligament [12,13,14]. MSCs have also been identified in the dental papilla located at the root apex of the developing permanent teeth and in the dental follicle of the developing tooth germs [15,16,17]. These subpopulations appear to share several qualities as MSCs, but their phenotypes and tissue regenerative potentials are not completely consistent [9,10,11]. Further investigations are required to define the differential phenotypes and tissue regenerative potentials in these subpopulations, particularly in MSCs associated with developing tooth germs. However, importantly, these dental MSCs can be obtained by minimally invasive procedures based on the clinical diagnosis of non-functional or pathogenic tissues in the oral cavity. In addition, they are usually discarded as medical waste after being removed from the oral cavity. Dental MSCs offer unique advantages for extensive medical research and applications than MSCs from other tissues.

Many studies have demonstrated the potential applications of dental MSCs, obtained from healthy subjects, in tissue regeneration and cell therapy. However, there are only a few studies highlighting the developmental modeling of tissue defects or genetic disorders. Despite the identification of the candidate genes, the exact mechanisms of many common genetic or rare congenital disorders remain elusive, and effective therapeutic strategies are lacking [18]. Human cellular models are essential for elucidating molecular pathologies, identifying specific therapeutic targets, and developing effective treatment options.

This review discusses the availability, limitations, and perspectives of dental-pulp-derived MSCs as human-disease-modeling systems currently being developed, along with our most recent findings.

## 2. Dental-Pulp-Derived Mesenchymal Stem Cells

The dental pulp is a highly vascularized connective tissue located in the center of the tooth. It is surrounded by mineralized hard tissue and comprises multiple cell types, including odontoblasts and undifferentiated progenitor cells [19]. The undifferentiated progenitor population also includes and segregates MSCs that show highly proliferative and multipotent capabilities in vitro and in vivo [20,21,22,23,24]. The dental pulp is a mesenchymal derivative of multipotent cranial neural crest cells that migrate to the first and second branchial arches during early embryonic development, indicating that MSCs in dental pulp are of neural crest origin [25,26,27,28,29]. Recently, MSCs were also detected in the apical papilla associated with the developing roots of permanent immature teeth called stem cells from the apical papilla (SCAP) [15,16]. SCAP has a dental papilla origin, similar to the dental pulp [30]. The physiological function of SCAP is to contribute to root development. They can also survive in immature permanent teeth with necrotic pulps to trigger their root repair, growth and regeneration through regenerative endodontic therapy [31,32,33]. Thus, developing tissue-derived SCAP are highly resistant to cell damage, and they might be a genetically and functionally early stage MSC population compared to the MSCs in the mature dental pulp [30,31,32,33]. Although they are cytologically categorized as MSCs, they also have neural crest-derived neurogenic potential, representing unique properties that differ from bone-marrow-derived mesenchymal stem cells (BM-MSCs) [6,34,35].

There are three main types of teeth in humans: deciduous, permanent, and supernumerary teeth, which share basic morphological structures and developmental pathways but partially differ at molecular levels [36,37]. MSCs isolated from the dental pulp of these teeth show highly proliferative properties and multipotency, although several differences have been observed at the molecular and cellular levels [20,21,22,23,24,38]. Interestingly, these cells transplanted into rodent nerve injury models suppressed excessive inflammation and apoptosis that interfere with tissue repair while differentiating into mature oligodendrocytes to promote neuro-regeneration [39,40]. These cells secreted monocyte chemoattractant protein-1 (MCP-1) and sialic acid-binding Ig-like lectin-9 (ED-Siglec-9) to induce anti-inflammatory M2-like macrophages through paracrine action, suggesting the possible mechanisms of the neuro-regeneration by these cells [41]. Among the properties of dental-pulp-derived MSCs, the immunomodulatory function has also been shown in BM-MSCs. In a mouse model of bisphosphonate-related osteonecrosis of the jaw, necrotic bone lesion might be associated with suppressing regulatory T cells and activating IL-17-producing T helper cells, and the imbalance between these two T helper subsets could be improved by administration of BM-MSCs [42]. However, the BM-MCs-mediated bone formation in recipient mice is reportedly inhibited by recipient proinflammatory T helper cells via interferon γ-mediated runt-related transcription factor 2 (RUNX2) pathway inhibition and tumor necrosis factor α (TNFα)-mediated apoptosis [43]. Although these BM-MC studies present some important factors to consider for the application to cell-based therapy, the functions of dental-pulp-derived MSCs have opened up new possibilities for dental pulp as a stem cell source for tissue engineering.

This section provides a basic understanding of the unique MSCs required for discussing the availability of these cells as disease models for genetic and congenital disorders. In the sections below, the two major MSCs derived from deciduous and permanent teeth, generally referred to as stem cells from human exfoliated deciduous teeth (SHEDs) and dental pulp stem cells (DPSCs), respectively, are briefly reviewed.

### 2.1. SHEDs

In healthy humans, 20 deciduous teeth develop, and they sequentially exfoliate and are replaced by permanent successors according to developmentally determined timing at approximately 6 to 12 years of age [37,44]. Deciduous teeth either spontaneously exfoliate upon being extruded by the erupting permanent successors, or alternatively, they are surgically extracted before interfering with erupting permanent successors. In both situations, exfoliated deciduous teeth can be collected via a minimally invasive procedure. These exfoliated or extracted deciduous teeth entail residual pulp tissues containing a sufficient number of MSCs, first described as SHEDs in 2003 [12]. SHEDs express mesenchymal stem cell markers, including STRO-1 and embryonic stem cell markers, and lose hematopoietic markers, including CD45 and CD11b/c [12]. The multipotency of SHEDs is demonstrated by the adipose, osteogenic, and chondrogenic lineages, which are shared with BM-MSCs [12]. In addition, the expression of nestin, a neural pro-genitor marker, is important evidence supporting the neural crest origin and neurogenic potential of SHED [12,45]. This is further highlighted by differentiation into dopaminergic neurons (DA neurons), a neuron subtype involved in brain function, including cognitive and motor function [46]. Transplantation of SHED-derived DA neurons reportedly alleviates Parkinson’s symptoms in model animals, suggesting SHEDs as a promising source of cell replacement therapy for the disease [47,48,49,50].

There are several considerations for establishing patient-derived SHEDs to model human diseases. Given that an individual develops 20 deciduous teeth, a total of 20 deciduous teeth can theoretically be obtained from a child suffering from a genetic disorder without affecting tooth development. This indicates 20 opportunities to collect SHEDs from a child. This is an advantage of deciduous teeth as a dental stem cell source over wisdom teeth, the most common source among adults; there are at most four sets in an individual. However, if craniofacial defects are associated with genetic disorders, the available number of deciduous teeth may be limited [51]. More importantly, premature decay or loss of deciduous teeth is often observed in uncontrolled oral pathogenic infections in children with severe systemic symptoms [52,53]. These are significant disadvantages for establishing patient-derived SHEDs as disease models. Thus, professional dental management is essential for the oral health and quality of life of children with severe systemic phenotypes but also critical for establishing reliable patient-derived SHED modeling of genetic disorders.

### 2.2. DPSCs

MSCs of the dental pulp of human permanent teeth can be expanded as single colonies in vitro to form a dentin-pulp complex in vivo, which was initially reported as postnatal DPSCs in 2000, prior to SHEDs [43]. DPSCs show multipotency, including osteogenic, adipogenic, and neurogenic differentiation, similar to SHEDs. Based on the high differentiation plasticities, DPSCs are currently being explored as a unique stem cell source for cell-based medicine, tissue regeneration, and tissue engineering for various diseases [20,21,22,23,24,39,40,41].

DPSCs are obtained from the dental pulp of permanent teeth, including surgically extracted wisdom teeth not used for occlusion primarily in mature permanent dentition. Many human genetic or congenital disorders develop from fetal or early postnatal stages and require early diagnosis and therapeutic intervention. To model these diseases, DPSCs, which are most likely to be obtained in the adult stage, may have a disadvantage compared to SHEDs, typically obtained from children as early as 6 years of age. However, DPSCs will be available as an alternative to SHEDs if SHEDs are unavailable.

The next section outlines the methods for inducing the differentiation of SHEDs and DPSCs into DA neurons, osteoblasts, and chondrocytes in our laboratory (Figure 1).

Teeth are collected from patients via minimally invasive procedures (STEP 1). The dental pulp is dissected from each tooth, and cells are dispersed by enzymatic treatment (STEP 2), followed by primary culture and storage in liquid nitrogen before use (STEP 3). The pathological analysis is performed by differentiating them into specific cell types suitable for the genotypes and phenotypes of the patients (STEP 4).

## 3. Collection, Culture, and Differentiation of SHEDs and DPSCs

### 3.1. Collection and Culture of SHEDs and DPSCs

Since they are present in the dental pulp, the affordable collection of SHEDs and DPSCs from affected patients depends on the availability, accessibility, and stable supply of teeth that retain viable pulp tissues. Our laboratory collaborates directly with the Department of Pediatric Dentistry and Special Needs Dentistry at Kyushu University Hospital. This department specializes in the comprehensive dental management of children suffering from various genetic and congenital diseases, collaborating with the Department of Pediatrics at Kyushu University Hospital. In many severely affected cases, oral function and development management begins even before a deciduous tooth eruption. These systems ensure both clinical care of deciduous teeth and their collection at the exfoliation stage. The exfoliated teeth collected in the clinic are immediately placed in a bottle containing standard culture media and promptly sent to the laboratory. In the laboratory, the dental pulp is dissected aseptically from the tooth, and SHEDs or DPSCs are isolated ac-cording to standard protocols and stored in a liquid nitrogen tank before analysis. All clinical information, including the phenotype and/or genotype of each stem cell founder, is strictly governed at Kyushu University Hospital and shared between departments. This information is essential for the clinical translation of the cellular phenotype.

### 3.2. Differentiation into DA Neurons

Although these are not yet completely established, different stem cell types and neural progenitor cells have been differentiated into DA neurons [54]. Differentiation of SHEDs and DPSCs into DA neurons has been confirmed by several research groups [47,48,49,50,55]. In our laboratory, both SHEDs and DPSCs are differentiated into DA neurons according to the method reported by Fujii et al. [49]. In this method, cells are cultured for 2 days in Dulbecco’s modified Eagle’s medium supplemented with N2, epidermal growth factor, and basic fibroblast growth factor in the first step to promote neuronal differentiation. In the second step, the culture medium was replaced with an induction medium composed of Neurobasal medium, B27 supplement, dibutyryl cyclic AMP (db-cAMP), 3-isobutyl-1-methylxanthine (IBMX), and ascorbic acid for an additional 5 days of culture.

However, unlike the previous method, the addition of brain-derived neurotrophic factor (BDNF) has been modified. Of note, Fujii et al. demonstrated a positive effect of BDNF on the survival and maturation of DA neurons. Genetic defects have been shown in the BDNF autocrine or paracrine pathways in many neurological disorders, suggesting their pathological roles [56,57,58,59,60]. Thus, BDNF supplementation in media may mask intrinsic defects of BDNF pathways in patient-derived SHEDs. In our laboratory, a protocol without BDNF supplementation is tested, and exogenous BDNF is added to the media to examine potential defects in BDNF pathways depending on the type of disease. Under conditions without BDNF, both SHEDs and DPSCs from healthy subjects are differentiated into DA neurons expressing tyrosine hydroxylase (TH), one of the DA neuron maturation markers, with a survival rate of 70–80%. Besides, DA neuron differentiation efficiency can be further evaluated by the expression of molecules, including NURR1 encoding the nuclear receptor-related 1 protein and PITX3 encoding the pituitary homeobox 3, which are essential transcription factors for the specificity of midbrain DA neurons [61,62,63].

### 3.3. Osteogenic and Chondrogenic Differentiations

Differentiation of SHEDs and DPSCs into osteoblasts and chondrocytes has been reported by many researchers [64,65]. For osteoblast differentiation, we use a protocol that combines β-glycerophosphate, ascorbic acid, and dexamethasone [66]. Phenotypes of osteoblasts are evaluated depending on the specific defects of diseases, but mainly by expression of differentiation markers, including genes encoding the alkaline phosphatase (ALP), the type I collagen, and the osteocalcin, expression of RUNX2, a master transcription factor for osteoblast differentiation, and mineralization with Alizarin red staining.

Chondrocytes are differentiated in high-density micromass cultures in the presence of transforming growth factor beta-3, dexamethasone, and insulin-transferrin-selenium mixture [67]. This method shows high cell viability after differentiation, is simpler than pellet culture, and is suitable for multiple samples. The chondrocyte phenotype is mainly evaluated by the expression of essential transcription factors for chondrocyte differentiation, including SRY-box 9 (SOX9) [68], and Alcian blue staining, which detects proteoglycan containing chondrocytes.

## 4. SHEDs and DPSCs Derived from Children with Genetic Disorders

Given their multipotency and disease-related genetic background, patient-derived SHEDs and DPSCs can be useful as a cellular model to elucidate genetic disease pathology. We have utilized the ability of SHEDs and DPSCs to differentiate into DA neurons, osteoblasts, and chondrocytes to elucidate the pathological mechanisms of several genetic disorders characterized by midbrain dopaminergic system and skeletal defects. The midbrain dopaminergic system is a diffuse modulatory system that plays a vital role in multiple brain functions, including motor, cognition, and rewards [46]. Altered differentiation, development, and function of DA neurons have been shown in many neuropsychiatric disorders, although the exact mechanisms remain undetermined [46]. Genetic defects may also affect bone metabolism, resulting in specific skeletal phenotypes [68]. Skeletal development progresses with well-organized temporal and spatial regulation by multiple cell lineages, including chondrocytes, osteoblasts, and osteoclasts [69]. This section describes the availability of patient-derived SHEDs and DPSCs to elucidate the pathophysiology of diseases caused by monogenetic variants or mtDNA mutations and polygenic neurodevelopmental disorders.

### 4.1. Rett Syndrome

Rett syndrome (RTT) is a progressive neurodevelopmental disorder primarily occurring in women [70]. Patients develop normally until 6–18 months of age, whereas brain growth retardation onsets around 2 years of age. Neurological features include stereotypic hand wringing or washing movements, ataxia, breath-holding associated with autonomic abnormalities, seizure, and autism-like symptoms. The most common causative gene for RTT is MECP2, which encodes methylated CpG-binding protein 2 (MeCP2) located on the X chromosome, and mutant MECP2 heterozygous males are generally lethal [71]. MeCP2 is a multifunctional transcription regulator that is enriched in the brain, and its loss of function disrupts the expression of a large number of genes required for brain development [72]. However, the molecular mechanisms underlying RTT neuropathology are not yet fully understood.

One of the main reasons that make it difficult to understand RTT neuropathology is the complicated actions of gene regulation of MeCP2. Transcriptional repression that selectively binds to methylated CpG and interacts with histone deacetylase (HDAC) and suppressor interacting 3a (Sin3a) is consistent with DNA methylation, which is generally involved in epigenetic gene repression [73,74]. However, extensive analysis of gene expression profiles in MeCP2 mutant mice’s brains has shown that gene expression is bidirectionally altered, depending on the differential brain regions, by MecP2 deficiency [75,76,77]. cAMP response element-binding protein (CREB), which can bind to a 5-hydroxymethylcytosine abundant region in transcriptionally active genes, has also been identified as a binding partner for the transcriptional activation by MeCP2 [76,78]. Thus, MeCP2 may have brain-region-specific functions to differentially regulate gene expression, although the exact mechanisms remain unresolved.

An additional reason for the complex neuropathology of RTT is that MECP2 is affected by X chromosome inactivation (XCI) [79]. XCI is a developmental process that silences most genes on one X chromosome in females, resulting in X-linked gene dosage compensation between females and males [80]. XCI begins during the early embryonic stage, and the maternal or paternal X chromosomes are randomly inactivated in placental mammals, including humans. Random XCI roughly results in half of the cells with an inactive maternal X chromosome and the other half of the cells with an inactive paternal chromosome, a mosaic individual. Thus, RTT females have a mosaic of wild-type cells and MECP2 mutant cells. These ratios are theoretically 50:50 under random XCI; however, skewed ratios have been identified in some RTT individuals, related to the heterogeneous phenotypes in RTT [81,82]. This should be considered when analyzing RTT patient-derived tissues and stem cells.

Although there are still many unresolved questions in RTT neuropathology, previous studies have suggested pathogenic elements, including biogenic amines and mitochondrial functions [83,84]. Dopamine is a biogenic amine associated with involuntary movements and autonomic dysfunction in RTT symptoms. Patients and MeCP2-deficient models in RTT have shown altered expression of many genes encoding proteins that control mitochondrial morphology and function [84,85]. Further, biogenic amines and mitochondria have also been proposed as potential therapeutic targets for RTT [83,84]. Given that MeCP2 has specific functions in different brain regions composed of different neuron subtypes [75,76,77], the pathological association between monoaminergic neurons and mitochondria should be an important element in understanding the neuropathology of RTT. We investigated the involvement of mitochondria in the differentiation and maturation of DA neurons with MeCP2 deficiency, using SHEDs obtained from a child with RTT, with a large deletion that included exons 3 and 4 of MECP2 [86].

It is essential to separate wild-type SHEDs and MeCP2-deficient SHEDs, as RTT peripheral tissue-derived SHEDs are a mosaic of these cells. In a previous study, wild-type SHEDs expressing MeCP2 and mutant SHEDs lacking MeCP2 were isolated by limiting dilution in combination with immunofluorescent staining of cells with anti-MeCP2 antibody to distinguish both cells [86]. The isolated wild-type SHEDs provided an isogenic control with the same genetic background as the MeCP2-negative SHEDs, compared to the control SHEDs isolated from a healthy individual.

In the above study, it is also important to consider whether SHEDs have a stable post-XCI status. Although the mechanism of XCI has not been fully elucidated in humans, it most likely starts in the early blastocyst and is completed at post-implantation [87]. SHEDs have a cranial neural crest cell origin and migrate from the tip of the neural fold to the first and second pharyngeal arches during neural tube formation, which occurs more than 2 weeks after implantation [88]. Thus, SHEDs are most likely to show a stable post-XCI status, compared to induced pluripotent stem cells (iPSCs) established from RTT patients [89].

DA neurons differentiated from both SHEDs showed comparable levels of immuno-fluorescent staining of TH, suggesting no critical defects of neuro-differentiation potential in the mutant SHEDs. However, MeCP2-deficient DA neurons showed a significant reduction in the length and number of branches of neurites, mitochondrial membrane potential (MMP), ATP levels, and mitochondrial distribution to neurites, compared to isogenic control DA neurons. These results suggest that MeCP2 deficiency suppresses DA neuron development associated with mitochondrial dysfunction. Furthermore, the expression of DRP1, an essential mitochondrial fission factor involved in mitochondrial recruitment to developing neurites [90], was significantly reduced in MeCP2-deficient DA neurons. This suggests that DRP1 is a potential target gene downstream of MeCP2.

Thus, RTT-patient-derived SHEDs can contribute to elucidating the molecular mechanisms of midbrain dopaminergic defects that may be associated with RTT neuropathology.

### 4.2. Metatropic Dysplasia

Metatropic dysplasia (MD) is the most severe form of the rare autosomal dominant inherited skeletal dysplasia caused by a heterozygous mutation in TRPV4, which encodes the transient receptor potential vanilloid 4 [91,92]. The main features of MD are severe platyspondyly, dumbbell-like deformity of the long tubular bones, and progressive kyphoscoliosis. MD shows a broad spectrum of severity, from non-lethal to lethal [91,92]. In non-lethal MD, short limbs are observed at birth due to a severe defect of long bone development followed by progressive kyphoscoliosis after birth, resulting in a reversal of body proportions (i.e., from ‘short limbs’ to ‘short trunk’) [92]. The underlying pathology of TRPV4-related diseases, including MD, is not yet fully understood, mainly because of the rarity of cases, widespread TRPV4 mutations, heterogeneous skeletal phenotypes, and homo- or hetero-tetramer formation for channel function [92,93,94,95].

TRPV4 is a 6-transmembrane Ca^2+^ permeable ion channel ubiquitously expressed in the cell membrane and is polymodally activated by noxious stimuli, including heat, osmotic pressure, and mechanical stress [96]. Disease-related TRPV4 mutations are predominantly single missense mutations leading to constitutive channel activation [91,92,93,94,95]. Several studies have been reported to explain the pathology of skeletal dysplasia caused by TRPV4 mutations. Histopathological examination of the bones of patients with lethal MD suggested dysregulation of endochondral ossification characterized by extensive nodular proliferation of immature cartilage at the epiphysis and disorganized columnar formation by prehypertrophic and hypertrophic chondrocytes [91]. Analysis of transgenic mice expressing mouse mutant Trpv4 (R594H), the same mutation as one of human TRPV4-related skeletal dysplasia, under the control of the cartilage-specific Col2a1 promoter, showed lethal defects of endochondral ossification similar to MD [97].

Several cell models have also been analyzed. Ectopic overexpression of disease-related mutant TRPV4 in cell lines has been reported to increase intracellular Ca^2+^ at the basal levels and by various agonistic stimuli, indicating hyperactivity of mutant TRPV4 channels [91]. Chondrogenic differentiation of iPSCs established from a patient with lethal MD carrying TRPV4 p.I604M showed abnormal expression of differentiation markers including SOX9 and irregular staining pattern of Alcian blue, compared to the healthy subject-derived cultures [98]. Primary chondrocytes from patients with non-lethal MD caused by mutations in p.G800D and p.P799L showed elevated intracellular Ca^2+^ entry in response to various agonists, suggesting TRPV4-induced Ca^2+^ dysregulation in chondrocytes [99]. Functional screening using a cDNA library derived from the murine chondrogenic cell line, ATDC5, identified TRPV4 as a gene that induces SOX9 expression, suggesting an essential role of TRPV4 in chondrogenic differentiation [100]. These findings suggest that pathological mutations of TRPV4 disturb endochondral ossification with defective chondrocyte differentiation through hyperactivation of the Ca^2+^ channel function.

Endochondral ossification begins with the aggregation of mesenchymal progenitor cells and progresses with multiple cells, including chondrocytes, osteoblasts, and osteoclasts, all of which express TRPV4 [69,93]. Therefore, it is likely that chondrocytes and other cell lineages may be involved in MD bone phenotype. However, cell type-specific pathology has not yet been well examined. We established DPSCs from a permanent tooth obtained through dental treatment for a 14-year-old boy diagnosed with MD to analyze them as a mutant mesenchymal stem cell model capable of being induced to differentiate into chondrocytes and osteoblasts [101,102].

DPSCs obtained from this boy (MD-DPSCs) were first sequenced for TRPV4, and a novel heterozygous single nucleotide mutation c.1855C>T was identified, which was predicted to be a missense mutation (p.L619F) in the putative transmembrane segment 5 of TRPV4 [101]. To further analyze the function of the mutated TRPV4, this missense mutation was repaired using the CRISPR/Cas9 system to isolate isogenic control DPSCs (Ctrl-DPSCs) with the same genetic background as the mutant DPSCs except for TRPV4. The intracellular Ca^2+^ levels of both DPSCs were comparable at the basal level. However, MD-DPSCs showed significantly higher intracellular Ca^2+^ levels than did Ctrl-DPSCs when stimulated with a specific agonist of TRPV4, 4α-phorbol 12,13-didecanoate (4αPDD). Upregulation of SOX9 and higher staining intensity of Alcian blue were observed in mutant chondrocytes differentiated in the presence of 4αPDD, compared to those of isogenic controls, suggesting accelerated chondrocyte differentiation from MD-DPSCs through enhanced Ca^2+^ signaling via mutant TRPV4.

Meanwhile, osteoblasts differentiated from MD-DPSCs showed increased calcification, as indicated by higher Alizarin red staining intensity, accompanied by increased expression of Runx2 and osteocalcin [102]. Expression and nuclear translocation of nuclear factor-activated T cells c1 (NFATc1) were also enhanced in the mutant osteoblasts, consistent with higher intracellular Ca^2+^ levels than isogenic controls. Cytosolic NFATc1 translocates to the nucleus after dephosphorylation by calcineurin, a Ca^2+^-dependent protein phosphatase, and functions as a transcription factor that promotes osteoblast differentiation. These results suggest accelerated differentiation of MD-DPSC into osteoblasts through enhanced Ca^2+^-dependent signaling.

Thus, a novel missense mutation c.1855 C>T in TRPV4 is a gain-of-function mutation that enhances intracellular Ca^2+^ signaling in MSCs and upregulates Ca^2+^-responsive pathways to accelerate both chondrocyte and osteoblast differentiation. These data suggest the possible mechanisms involved in the developmental defects of bone in the examined MD and indicate the potential of patient-derived DPSCs as a useful cellular model for the analysis of the pathological mechanisms of MD.

### 4.3. Leigh Syndrome

Mitochondria play multiple vital roles in cellular physiology, such as energy production by oxidative phosphorylation, metabolism of various substances, redox control, calcium homeostasis, and apoptosis regulation [103]. Leigh syndrome (LS) is primarily a functional defect in oxidative phosphorylation caused by mutations in either the nuclear genome or mtDNA that encode the mitochondrial respiratory chain complexes [104,105]. LS can affect any tissue throughout the body; however, high ATP-demanding organs, including the brain, heart, and muscle, are particularly vulnerable, leading to more clinical manifestations [104,105]. The brain’s typical pathology is bilateral neurodegenerative lesions from the basal ganglia to the brain stem [106].

A comprehensive understanding of the pathology and effective therapeutic targets in LS has not yet been established. The first reason is the heteroplasmy of mtDNA [107]. mtDNA is a 16.5 kb double-stranded circular molecule that encodes 13 subunits of the respiratory chain complexes, 2 rRNAs, and 22 tRNAs, ranging from hundreds to thousands copies in a cell [107]. Each patient has both mutant and wild mtDNA copies (heteroplasmy), and the ratio between them may show variation between patients and even different tissues of a patient, resulting in the heterogeneous pathology and severity of LS carrying mtDNA mutations [104,105]. The second reason is the multiple functions of mitochondria are closely related to each other [103,108]. It is essential to determine both the defects of each mitochondrial element and their integrated results to understand LS genotype-phenotype correlation. The third reason is the establishment of disease models [109,110]. Several iPSCs have been successfully established from patients with LS carrying mtDNA mutations and differentiated into muscles and neurons for analysis [111,112]. However, reprogramming to establish iPSCs may affect the nuclear genome and mtDNA stability and the patient-specific heteroplasmic state associated with uneven mitochondrial segregation during reprogramming [113,114,115].

We established SHEDs derived from a 6-year-old boy diagnosed with LS (LS-SHEDs) caused by a G13513A mutation in the mtDNA encoding the ND5 subunit of respiratory chain complex I [116]. Patients with LS are predominantly characterized by neuropathology, but other organs, including bone, are also severely affected [117,118]. However, the direct effects and molecular mechanisms of mitochondrial dysfunction on bone health have not been extensively elucidated. We used LS-SHEDs as a cellular model and compared them to SHEDs derived from a typically developing child (Ctrl-SHEDs) to analyze the role of mitochondrial dysfunction in osteoblast differentiation and their potential therapeutic targets [116].

Our patient showed a 39% reduction in bone mineral density (0.381 g/cm^2^) compared to the age-matched standard value (0.581 g/cm^2^), along with typical neuropsychiatric symptoms of LS. PCR analysis of heteroplasmy revealed that 50% of the total mtDNA was mutated in LS-SHED, and no G13513A mutation was detected in the Ctrl-SHEDs. Com-pared to the Ctrl-SHEDs, the LS-SHEDs showed significantly lower MMP and ATP levels, suggesting the impaired oxidative phosphorylation associated with genetic defects in mitochondrial respiratory chain complex I. Similar mitochondrial dysfunctions were also detected in differentiated osteoblasts (LS-OBs), compared to control osteoblasts (Ctrl-OBs). Differentiation into LS-OBs was significantly impaired, as confirmed by the lower staining intensity with Alizarin red and lower expression of ALP. Furthermore, both intracellular and mitochondrial Ca^2+^ levels were reduced in LS-OBs. Intracellular Ca^2+^ is a second messenger required to activate RUNX2 through β-catenin [119,120,121,122]. Mitochondrial Ca^2+^ is an important source of Ca^2+^ for matrix vesicles required for bone mineralization [123,124,125]. The expression of Kir6.2, the component of ATP-sensitive potassium (KATP) channels, is highly upregulated during osteogenic differentiation to regulate intracellular Ca^2+^ levels [126]. Therefore, it is likely that the impaired oxidative phosphorylation caused by m.13513G>A disturbs mitochondrial function and Ca^2+^ homeostasis required for osteogenic differentiation, which is involved in the bone phenotypes in LS.

Mitochondrial dysfunction is reportedly ameliorated by mitochondrial biogenesis [127]. Therefore, we focused on mitochondrial biogenesis as a potential therapeutic target for bone defects in LS [128]. PGC-1α is a master regulator of mitochondrial biogenesis [129]. We examined BZF to promote mitochondrial biogenesis because BZF has been re-ported to improve bone metabolism and osteoporosis through PGC-1α upregulation [130,131]. Possible mechanisms of BZF are to stimulate the peroxisome proliferator response element in the PGC-1α gene promoter [127,131,132]. We found that BZF supple-mentation significantly improved both osteoblast differentiation and mineralization combined with ALP upregulation and β-catenin activation in LS-derived cultures. BZF-treated LS-OBs also showed increased mitochondrial network formation and mitochondrial con-tent, associated with PGC-1α upregulation and Drp1 downregulation. ATP and mitochondrial Ca^2+^ levels, but not MMP, were increased in BZF-supplemented LS-OBs. These results suggest that PGC-1α-mediated mitochondrial biogenesis may be a therapeutic tar-get to improve bone defects with impaired oxidative phosphorylation due to m.13513G>A. BZF may be a potential candidate for pharmacological intervention of LS bone defects.

### 4.4. Autism Spectrum Disorder

Autism spectrum disorders (ASD) are neurodevelopmental disorders characterized by social interaction and communication deficits and limited repetitive interests and behaviors [133]. ASD is a complex multifactorial disease with highly heterogeneous clinical manifestations, except for syndromic ASD caused by specific gene mutations, including RTT, fragile X syndrome, and Angelman syndrome [134,135,136,137]. Investigations of postmortem human brains, metabolites of blood samples, model mice, and genome-wide association studies have shown many genetic and environmental risk factors associated with ASD [134,135,136,137,138,139,140]. Based on these findings, many hypotheses have been proposed to explain the underlying neuropathology of ASD, including mitochondrial dysfunction and the dopamine hypothesis [141,142,143,144], which are highlighted in our study.

As described in other sections, mitochondria have several functions, including energy production, redox regulation, and Ca^2+^ homeostasis, all of which are essential for brain development and function, indicating the extreme vulnerability of the brain to mitochondrial dysfunction [103,108]. Mitochondrial dysfunction has been shown in many neuropsychiatric disorders, and accumulating evidence also shows an association between ASD and mitochondrial dysfunction [143,144]. Studies of postmortem brains of individuals with ASD have shown an apparent reduction in the protein levels or activity of the mitochondrial respiratory chain complex in various brain regions [145]. Biochemical analysis using peripheral blood or tissues of patients with ASD has reported metabolic alterations of lactic acid that reflect oxidative phosphorylation defects [139,145]. These results strongly suggest the association of unidentified genetic or epigenetic pathways with mitochondrial dysfunction in the ASD brain.

However, the dopamine hypothesis proposes that the core symptoms of ASD may be associated with dysfunction of the midbrain dopaminergic system [142]. Midbrain DA neurons, with their cell bodies in the midbrain, project their neurites widely throughout the brain. They contribute to multiple functions such as cognitive function, reward system, and motor control, and are further classified into functionally different subtypes [46]. The mesocorticolimbic system projects from the ventral tegmental area to the prefrontal cortex and the ventral striatum nucleus accumbens and is involved in reward and motivation [46]. Dysfunction in this pathway reduces social reward and motivation and may contribute to social interaction deficits in ASD [133,142]. The nigrostriatal system projects from the substantia nigra to the dorsal striatum and controls goal-directed motor behavior [46]. Dysfunction in this pathway may involve limited, purposeless, and stereotyped pat-terns of behavior [133,142]. Altered dopaminergic signals associated with decreased DA levels and DA receptor gene variations in the brain have been suggested in ASD [146]. Pharmacological modulation of the brain DA levels has been shown to alleviate ASD symptoms [142,147].

Thus, many results have been independently presented to support the pathological involvement of mitochondria and DA systems in ASD; however, few studies have shown a direct association in ASD neurobiology. The limitations in ASD research include the brain anatomical and functional complexity and the technical and ethical issues of sampling and analyzing affected individuals’ brains. We used SHEDs derived from children with ASD to examine the neurobiology of ASD, highlighting the early developmental defects of DA neurons and mitochondria.

SHEDs from three children diagnosed with ASD and three age-matched healthy children as controls were differentiated into DA neurons [148]. ASD-derived SHEDs differentiated into DA neurons (ASD-DNs) without severe defects, as supported by comparable levels of NURR1, PITX3, and TH expression compared to control DA neurons (Ctrl-DNs). Morphological analysis showed that both neurite length and branching were significantly reduced in ASD-DNs compared to Ctrl-DNs. MMP, intracellular ATP levels, and intracellular Ca^2+^ levels were also significantly reduced in ASD-DNs. In addition, the mitochondrial content per cell and the number of mitochondria-containing neurites were significantly reduced in ASD-DNs. These results suggest that defects in mitochondrial oxidative phosphorylation, biogenesis, and recruitment to neurites were associated with impaired neurite development in ASD-DNs. BDNF was supplemented in media; however, no clear improvement of these defects was observed in ASD-DNs.

Further analysis is required to determine the molecular pathology of DA neurons in ASD, including identifying molecules involved in the mitochondrial dysfunction and un-responsiveness to BDNF. However, this study demonstrated that SHEDs derived from children with ASD are a useful cellular model for analyzing defects of DA neurons and mitochondria involved in the development of ASD pathology.

## 5. Limitations and Perspectives

To date, many disease models have been established to understand the underlying pathology of human diseases and develop effective treatments; however, they have not yet fully covered human diseases and have several limitations.

Patient-derived iPSCs are excellent stem cell models carrying pathogenic mutations [149]. Many protocols and techniques have been proposed for in vitro 2D and 3D organoid cultures to obtain virtually any type of differentiated cell with disease-specific defects [150]. However, there are several concerns regarding iPSCs for disease-modeling. First, the experimental procedure is complicated, and the culturing cost is high. Second, reprogramming to induce pluripotency may affect genetic stability. For example, some genes introduced for reprogramming may add new mutations to the patient’s genome, generating tumor cells in cultures [113]. These reprogramming procedures may also affect the copy number of mitochondrial DNA (mtDNA), altering the heteroplasmic ratios associated with the phenotype severity of mitochondrial diseases, which are commonly intractable [114]. Reprogramming may also affect the epigenetic status, closely related to disease pathogenesis [115]. Altogether, genomic instability and epigenetic alterations may result in additional defects in patient-derived iPSCs, challenging clinical translation.

Patient-derived SHEDs and DPSCs have also many limitations and considerations as disease-specific cellular models. First, dental pulp-derived stem cells show high proliferative capacity; however, they do not possess infinite proliferative capacity, unlike iPSCs and immortalized cell lines. The number of passages should be carefully managed in each experiment to avoid artificial effects associated with cellular aging due to in-creased passages. Second, SHEDs and DPSCs have only limited plasticity than other pluripotent stem cell models, although they can differentiate into many different cell types. In particular, to model neurodevelopmental disorders, it is also necessary to investigate defects in excitatory and inhibitory neurons and other types of monoaminergic neurons. Currently, as the technology has not been established to induce differentiation into these neurons, SHEDs and DPSCs are limited to analysis focusing on the dysfunction of the midbrain DA system based on the patient’s phenotype. Third, DA neurons differentiated from SHEDs and DPSCs show difficulty in long-term survival relative to iPSC-derived neurons [151]. Therefore, there are many limitations in reproducing the developmental process of DA neurons until full maturation. Analysis of DA neuron pathology should focus on defects in the early developmental stage of DA neurons associated with neuro-developmental disorders. Reduced oxygen pressure may be an alternative protocol for long-term culture to minimize oxidative stress and more precisely mimic the physiological oxidative condition in vivo [49].

Although many limitations remain to be resolved, patient-derived SHEDs and DPSCs are reportedly useful for modeling some neuropsychiatric genetic disorders. Urraca et al. analyzed the detailed gene expression profile using neurons differentiated from SHEDs in children with copy number variation in 15q regions overlapping between ASD and Angelman syndrome [152]. These SHED-derived neurons showed insufficient silencing of imprinting genes due to immaturity, and cultures contained non-neuronal glial lineage cells. Although a careful interpretation of the results is required, patient-derived SHEDs are useful for screening and identifying genes with altered expression levels before and after neuronal differentiation. Altered gene expression profiles are associated not only with gene sequence variants but also with epigenetic marks [153]. In a global analysis of DNA methylation profiles, SHEDs reflected the in vivo methylation patterns found in the placenta and inner cell mass than iPSCs, in which DNA methylation status becomes unstable during reprogramming [154]. These findings demonstrate the potential benefits of SHEDs in investigating the epigenetic pathology associated with human diseases. Patient-derived SHEDs, along with disease models of specific genetic disorders, may also contribute to the early intervention by personalized medicine based on differential disease susceptibility associated with differential genetic or epigenetic variants among individuals affected by complex polygenic diseases represented by ASD.

## 6. Concluding Remark

This review described the current understanding of the fundamental properties of cranial neural crest-derived MSCs in the dental pulp of human deciduous and permanent teeth. Apart from being an excellent source of cell-based medicine and tissue regeneration, their applications and limitations as disease modeling systems are explained through several examples of RTT and MD caused by single gene mutation, LS caused by mtDNA mutation, and ASD categorized into polygenic diseases. We propose that these unique MSCs can be an alternative for a disease-specific cellular model that can at least partially complement other stem cell models, such as iPSCs.

## Figures and Tables

**Figure 1 ijms-22-02269-f001:**
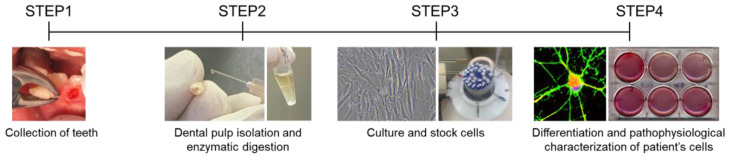
Workflow for the preparation of stem cells from human exfoliated deciduous teeth (SHEDs) and dental pulp stem cells (DPSCs) for pathological analyses.

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
