# Peer review of "Dental Pulp-Derived Mesenchymal Stem Cells for Modeling Genetic Disorders"

_ijms, 2021, doi:10.3390/ijms22052269_

Round 1

Reviewer 1 Report

The review article is very well written and of broad interest. The abstract is very good. The introduction however does not sit particularly well with the rest of the text as iPSCs are not really an alternative source (yet). The authors could instead start with MSCs from other sources as model systens and incorporate the iPSC section from the introduction into the limitations paragraph at the end of the review to discuss iPSCs in a single paragraph.

There are also some typos:

line 69: bone marrow

line 129: available

line 210: combines

line 221: containing

Author Response

We greatly appreciate the reviewer’s suggestion. We have provided our point-by-point responses to the reviewer’s comments below. All revisions requested by reviewers are marked in yellow in the revised manuscript.

1. We have revised the Introduction as suggested. The revised Introduction starts with a discussion on MSCs from other sources. We have also cited additional references in this section. The discussion on iPSCs was moved to the second paragraph of the limitation and perspectives section.

Lines 34–73 in “Introduction,” and Lines 547–564 in “Limitation and perspectives”.

2. Typos have been corrected.

Line 92: bone marrow

Line 148: available

Line 231: combines

Line 242: containing

In addition, several instances of misinformation and typos were also corrected during manuscript revision. These corrections have been underlined in red in the revised manuscript.

Reviewer 2 Report

This narrative article is under the scope of this journal; the topic is relevant for readers, and this research deals with potentially significant knowledge to the field.

However, there are some concerns in the about the present manuscript:

A major concern of the article does not mention the SCAPs, like a font of stem cells for genetic study’s.

  • The Dental papilla, when the root is formed, is in the apical zone, called the papilla apical (stem cells from apical papilla - SCAPs) that remain until the apical closure of the root. These are stem cells more immatures than SHED or DPCS, for this reason, one great font of genetic information. And we can collect them from patients with immature teeth approximately around 20 years. Also these, cells can the survival of SCAPs at the infection of the tooth, (in REPs read these references (org/10.3390/APP9193942 and DOI: 10.1016/j.joen.2017.03.005).
  • The Isolation and Culture https://doi.org/10.3390/jfb9040074)
  • The authors must also describe the Stem Cell from Apical Papilla SCAPs.
  • references are not standardized. The titles of references have a different format, 
    the title of the article is written in capital letters at the beginning of words, others only in lower case. Also, the standardized format of presentation in the journal's name. Because names have written in a different format, one is not abbreviated, others are not.

Author Response

We greatly appreciate the reviewer’s suggestion. We have provided our point-by-point responses to the reviewer’s comments below. All revisions requested by reviewers are marked in yellow in the revised manuscript.

1. The information on SCAPs was added to the revised manuscript.

Line 82–90 in “Dental pulp-derived mesenchymal stem cells”.

2. The reference section has been revised and the format standardized (highlighted in yellow).

In addition, several instances of misinformation and typos were also corrected during manuscript revision. These corrections have been underlined in red in the revised manuscript.

Round 2

Reviewer 2 Report

This research is under the scope of this journal; the topic is interesting for readers and this research deals with potentially significant knowledge to the field and an open new way for future studies.

The authors improved the quality of the manuscript after the reviewer's indications. Congratulations!!